# [^18^F]FDG Uptake in Non-Infected Endovascular Grafts: A Retrospective Study

**DOI:** 10.3390/diagnostics13030409

**Published:** 2023-01-23

**Authors:** Chiara Lauri, Alberto Signore, Giuseppe Campagna, Francesco Aloisi, Maurizio Taurino, Pasqualino Sirignano

**Affiliations:** 1Nuclear Medicine Unit, Department of Medical-Surgical Sciences and of Translational Medicine, Sant’Andrea Hospital, “Sapienza” University of Rome, 00161 Rome, Italy; 2Vascular Surgery Unit, Department of Clinical and Molecular Medicine, Sant’Andrea Hospital, “Sapienza” University of Rome, 00161 Rome, Italy; 3Vascular Surgery Unit, Sant’Andrea Hospital, Department of General and Specialistic Surgery, “Sapienza” University of Rome, 00161 Rome, Italy

**Keywords:** vascular graft infection, [^18^F]FDG PET/CT, endurant, aortic aneurysm

## Abstract

**Purpose:** After endovascular aneurysm repair (EVAR), an increased [^18^F]FDG uptake may be observed at PET/CT, being common to both vascular graft/endograft infection (VGEI) and sterile post-surgical inflammation. Increased non-specific metabolic activity, due to foreign body reaction, can persist for several years after surgery, thus complicating the interpretation of PET/CT studies. In this paper, we aimed to assess [^18^F]FDG distribution at different time-points after the implant of abdominal Endurant^®^ endografts in patients without suspicion of infection. **Methods:** We retrospectively evaluated [^18^F]FDG/CT in 16 oncological patients who underwent abdominal aortic aneurysm exclusion with Endurant^®^ grafts. Patients had no clinical suspicion of infection and were followed up for at least 24 months after scan. [^18^F]FDG PET/CT scans were interpreted using both visual and semi-quantitative analyses. **Results:** The time between the EVAR procedure and [^18^F]FDG PET/CT ranged between 1 and 36 months. All grafts showed mild and diffuse [^18^F]FDG uptake without a focal pattern. Mean values of SUVmax were 2.63 ± 0.48 (95% CI 2.38–2.88); for SUVmean 1.90 ± 0.33 (95% CI 1.72–2.08); for T/B ratios 1.43 ± 0.41 (95% CI 1.21–1.65). SUVmax and SUVmean were not correlated to the time elapsed from the procedure, but we observed a declining trend in T/B ratio over time. **Conclusions:** Endovascular implant of Endurant^®^ grafts does not cause a significant inflammatory reaction. The evidence of faint and diffuse [^18^F]FDG uptake along the graft can reliably exclude an infection, even in early post-procedural phases. Therefore, in patients with a low probability of VGEI, [^18^F]FDG PET/CT can also be performed immediately after EVAR.

## 1. Introduction

Endovascular abdominal and thoracic aneurysm repair (EVAR/TEVAR) are nowadays routinely performed endovascular procedures for the exclusion of abdominal aortic aneurysm (AAA) and thoracic aortic aneurysm (TAA). More than 75% of AAA and almost all TAA are, indeed, excluded by an endovascular approach, by EVAR and TEVAR, respectively [1,2,3]. Although these procedures are associated with lower morbidity and mortality rates, faster discharge, and fewer complications compared to open aneurysm repair, graft durability, the need for life-long postoperative surveillance, and reinterventions represent major concerns [4]. This higher rate of reintervention has been ascribed to a frequent occurrence of endoleak (mostly type II), limb occlusion, graft migration, and infection [5]. Although the great majority of such complications can be fixed by minimally invasive catheter-based reoperation, infection of the graft remains a serious complication [6,7].

Therefore, in the case of suspicion of vascular graft/endograft infection (VGEI), several diagnostic procedures should be applied. The diagnosis relies on a combination of clinical signs and symptoms, haemocultures, computed tomography angiography (CTA), radiolabelled autologous white-blood-cells (WBC) scintigraphy and positron emission tomography (PET) with 18F-fluorodeoxiglucose ([^18^F]FDG). The latter has been proposed by several authors as a reference diagnostic method, but it has variable diagnostic accuracy depending on the studied population and the interpretation criteria used [8,9,10,11,12].

One major limit of using [^18^F]FDG is the high false-positive rate due to sterile inflammation or post-surgical physiological inflammatory reaction. In 2014, Keidar et al. analyzed the [^18^F]FDG uptake in non-infected patients at different time-points from surgery (from 5 months to 16 years before the PET/CT study). Patients with both central and peripheral grafts made by different materials (Dacron, Gore-tex or native vein grafts) were analyzed [13]. However, no studies are available investigating the [^18^F]FDG uptake in Endurant^®^ after endovascular aortic repair, or in the early post-procedural phase.

Therefore, our aim was to retrospectively assess the pattern of [^18^F]FDG uptake in a group of cancer patients with abdominal Endurant^®^ grafts without any signs or symptoms of infection and studied at different time-points after EVAR.

## 2. Materials and Methods

### 2.1. Patients Population

We retrieved all oncologic patients studied by [^18^F]FDG PET/CT from January 2008 to October 2020 in the Nuclear Medicine Department of the Sant’Andrea Hospital of Rome. We selected subjects who underwent an EVAR procedure for AAA exclusion before the diagnosis of cancer. Only patients without suspected VGEI were finally included. The inclusion criteria were: adults over 18 years; patients with abdominal Endurant^®^ grafts from the manufacturer Medtronic^®^ (Santa Rosa, CA, USA); no clinical, radiological or biochemical suspicion of VGEI (normal inflammatory markers); follow-up of at least 24 months after PET/CT to confirm the absence of infection.

The exclusion criteria were: patients with different graft materials or different locations (e.g., peripheral grafts); patients with suspected VGEI according to MAGIC criteria [14]; follow-up less than 2 years; pluri-metastatic patients, with advanced tumor stage, or presenting with a large primary tumor close to the graft. This was decided to reduce the possible interference of hyper-metabolic neoplastic lesions on the evaluation of the [^18^F]FDG distribution along the graft.

We collected demographic data, laboratory test results, information on comorbidities, such as previous cardiac attack or heart failure, and risk factors (i.e., smoking, dyslipidemia, diabetes, hypertension), therapies, type of surgery and the time elapsed from the EVAR procedure.

### 2.2. Ethical Requirements

This study complied with the principles of the Declaration of Helsinki; patients gave their consent for the procedures and data collection and analysis. Due to the retrospective nature of the study, local ethical committees were only notified.

### 2.3. Preoperative Work-Up, Intervention, Postoperative Follow-Up

Indication for AAA repair was based primarily on aneurysm diameter; rate of growth more than 1 cm/year and aortic wall morphology were also considered Indications for EVAR were based on age, comorbidities, operators’ experience, and patient preferences [15]. Prior to the initial EVAR procedure, all patients underwent an extensive assessment, including clinical history reporting, physical examination, chest radiography, electrocardiography, pulmonary-function testing, transthoracic echocardiography, and laboratory testing. Patients also underwent CTA of the entire thoracic and abdominal aorta to evaluate the presence of other aortic lesions and to determine the feasibility of EVAR. CTA was performed with and without contrast medium during arterial and venous phases using a 1 mm slice thickness. All measurements were performed using a workstation with dedicated software and center lumen line reconstruction (OsiriX MD software version 12; PIXMEO, Bernex, Switzerland) [16].

EVAR procedures were all performed with Medtronic Endurant^®^ according to the manufacturer’s instructions for use (IFU); type of anesthesia, surgical or percutaneous access, and need for adjunctive intraoperative procedures (i.e., sac embolization and/or patent aortic branches embolization) were tailored for each patient (Figure 1).

After EVAR, the standard follow-up protocol included physical examination, duplex-ultrasound scan (DUS), and CT at 30 days (Figure 2). DUS was then performed at 3 and 6 months, at 1 year, and yearly thereafter. All patients underwent CT imaging 1 year after the initial procedure, without further CT examinations in the absence of complications detected by yearly DUS follow-up.

### 2.4. [^18^F]FDG PET/CT

Whole body scans were acquired approximately 1 h post injection (p.i.) of 2.5–4.5 MBq/kg of [^18^F]FDG. Hybrid PET/CT Gemini tomograph (Philips, The Netherlands), composed by a third-generation multi-slice spiral CT scanner and a full-ring PET scanner (bismuth germinate crystals), was used until July 2018. Hybrid PET/CT Biograph tomograph (Siemens, Germany), with multi-slice spiral CT scanner and a 5-rings PET scanner (lutetium oxyorthosilicate crystals), was used from September 2018. PET images were acquired for 2.5 min per bed position from head to mid-thigh. The low-dose CT scans were performed with the following parameters: 140 kV, 90 mA, 0.8/s tube rotation, 5 mm thickness. After correction for attenuation, PET images were automatically fused with CT images and displayed in maximum intensity projections (MIP) in axial, coronal and sagittal planes.

Three nuclear medicine physicians, with experience in imaging of infection and inflammation, evaluated the [^18^F]FDG PET/CT scans blind.

Qualitative assessment was performed using the five-point visual grading score suggested by Sah et al. [17]:

Score 1: [^18^F]FDG uptake comparable to the background;

Score 2: diffuse [^18^F]FDG uptake along the graft with mild intensity (<twice the blood pool activity in the ascending aorta);

Score 3: focal and mild [^18^F]FDG uptake or diffuse and strong (>twice the blood pool activity) [^18^F]FDG uptake;

Score 4: focal and intense [^18^F]FDG uptake (±diffuse);

Score 5: focal and intense [^18^F]FDG uptake associated with peri-graft signs of infection (e.g., fluid collections or abscesses).

According to this scale, score 1 and 2 were considered negative for infection, and scores 3–5 as positive.

For a semi-quantitative evaluation, the maximum and mean standardized uptake value (SUVmax and SUVmean) and target/background (T/B) ratios were analyzed as follows:

—drawing a circular region of interest (ROI) of 50 pixels on the site of the highest [^18^F]FDG uptake, covering the whole vessel for calculating SUVmax and SUVmean;

—drawing a 20 pixels circular ROI in the inner of the first tract of the descending aorta (between the 5th and 6th dorsal vertebra), for blood pool background. T/B ratios were calculated by dividing SUVmean of the target by SUVmean of the background.

## 3. Statistical Analysis

Continuous variables are shown as mean ± standard deviation (SD) and with the 95% confidence interval (CI). Categorical variables are expressed as absolute frequencies and percentages—n (%).

The correlations between SUVmax, SUVmean, T/B ratios and time from surgery were evaluated using Spearman’s coefficient ρ since the variables were not normally distributed. The Shapiro–Wilk test and Q–Q plot were used to verify the normality of the distribution of continuous variables.

A *p* value of <0.05 was considered statistically significant. Statistical analysis was performed using SAS version 9.4 and JMP PRO version 16 (SAS Institute, Cary, NC, USA).

## 4. Results

Sixteen oncologic patients (13 males, 3 females; mean age 71.81 ± 6.23 years), who previously underwent an EVAR procedure for the exclusion of AAA with Endurant^®^ stent graft, were included. None of the patients was clinically suspected for VGEI. Table 1 and Table 2 summarize the characteristics of the studied population. The EVAR procedure was successfully carried out in all patients; nine cases were treated under general anesthesia by bilateral femoral cut-down and four under local anesthesia by percutaneous common femoral artery access. Three type II endoleaks were detected at postoperative CTA in the absence of sac expansion. During the entire follow-up period, none of the enrolled patients required re-intervention.

All the 16 patients underwent [^18^F]FDG PET/CT investigation as part of the standard-of-care workup for their oncological disease that was diagnosed after EVAR. Five patients were studied for metabolic evaluation of a solitary lung nodule, occasionally detected during the pre-operative work-up for EVAR, four for restaging of hematological disease, three for a previous lung carcinoma, two for the restaging of gynecological neoplasm and two for staging of colon-rectal cancer. The time between EVAR and [^18^F]FDG PET/CT ranged between 1 and 36 months (11.75 ± 12.04 months). In detail, 5 out of 16 patients were studied in the first month; one patient was studied 4 months after EVAR; one patient was scanned at 6 months; one at 7 months; another one at 10 months; three patients at 12 months; two patients at 24 months; and 2 were studied 36 months after EVAR.

As expected, WBC count, CRP and ESR values showed a declining trend with increased time from surgery, but none of these inflammatory markers was significantly correlated with the time elapsed from surgery (Figure 3).

All PET/CT studies were correctly interpreted as negative by all the three readers, showing only mild and diffuse [^18^F]FDG uptake along the graft, without focal areas (Score 2) (Figure 4).

Mean values of SUVmax were 2.63 ± 0.48 (95% CI 2.38-2.88); mean values of SUVmean were 1.90 ± 0.33 (95% CI 1.72–2.08); and for the T/B ratio 1.43 ± 0.41 (95% CI 1.21–1.65). By analyzing the trend of semi-quantitative parameters according to time from EVAR, only the T/B ratios showed an inverse correlation over time, whereas neither SUVmax nor SUVmean were correlated with the time elapsed from the endovascular procedure (Figure 5).

Moreover, no significant differences were observed when comparing the [^18^F]FDG uptake in patients with and without each risk factor.

## 5. Discussion

To the best of our knowledge, this is the first study that has sought to retrospectively assess the [^18^F]FDG biodistribution in non-infected patients with Endurant^®^ endografts. Keidar and colleagues have previously addressed this topic. Their population included 43 patients with both central and peripheral grafts made with different materials (Gore-Tex, Dacron and native veins) and investigated using [^18^F]FDG PET/CT at different time-points from surgery. Their findings highlighted that diffuse [^18^F]FDG uptake, without focal patterns, may be observed up to 16 years after a surgical procedure resulting from a persistent sterile inflammation due to foreign body reaction. Although they included a larger population and studied different graft materials, their study did not analyze patients with Endurant^®^ graft in the early post-procedural phases [13]. Since surgical approaches and graft materials evolve rapidly, we strongly believe that a similar analysis should also be performed at the current time.

Despite the well-known lack of specificity of [^18^F]FDG in discriminating infection from sterile post-surgical inflammation, our retrospective study confirms that mild and diffuse [^18^F]FDG uptake along the graft represents a solid interpretation criterion to exclude an infection, even in patients studied within the first 4 months of surgery.

The possible added value of semi-quantitative analysis of [^18^F]FDG uptake over qualitative assessment, mainly with SUVmax, has been thoroughly investigated [18,19,20,21,22,23,24,25,26,27,28]. Several authors have also proposed different thresholds for SUVmax, but a reliable cut-off value able to distinguish an infected from an inflamed graft has not been identified. In our study, SUVmax, SUVmean and T/B ratios were always very low, even in patients studied one month after EVAR, when an increased uptake would be expected due to inflammation. Low metabolic activity and diffuse FDG distribution without focal patterns are, therefore, reliable tools to exclude an infection.

Another heavily debated issue is how to calculate T/B ratios. Some authors used SUVmax graft/SUVmax background, some SUVmax graft/SUVmean background, and others SUVmean graft/SUVmean background. Moreover, the choice of the reference tissue to be used as background (i.e., caval vein, liver, descending or abdominal aorta), differs amongst published studies, resulting in a wide heterogeneity of calculated thresholds [21,23,27,29,30,31]. In agreement with Keidar [13], we calculated T/B ratios by normalizing the SUVmean of the graft (target) for the SUVmean of the blood pool. Indeed, since we did not observe any focal uptake, but only faint and diffuse [^18^F]FDG distribution along the whole graft, we believe that SUVmax could be less representative than SUVmean, or even misleading,. However, further studies are warranted to definitively assess which should be the method of choice for calculation of T/B ratios.

The main limitation of the present study is the small sample size. This is due to the strict inclusion/exclusion criteria we adopted, but, on the other hand, we had a very homogeneous population. Nevertheless, with the increasing number of EVAR procedures being performed, and the growing number of oncological patients being studied with [^18^F]FDG PET/CT in the next years, we plan to study a larger population to eventually confirm these findings. Another limitation of the present study is the lack of follow-up PET/CT in our studied population. The possibility to evaluate the same patient at different time-points with [^18^F]FDG PET/CT would have been very informative and would have strengthened our findings. Nevertheless, we have longitudinal information for only two patients who showed, at follow-up PET/CT, very similar semi-quantitative values and identical [^18^F]FDG uptake patterns, compared to the basal [^18^F]FDG PET/CT study. We expect, in the near future, to re-evaluate these patients in order to provide more robust evidence.

The comparison between [^18^F]FDG PET/CT studies in infected and non-infected grafts also needs to be considered, aiming to define well-standardized interpretation criteria for [^18^F]FDG in VGEIs.

## 6. Conclusions

EVAR procedure with Endurant^®^ grafts does not cause a significant inflammatory reaction. Non-infected vascular grafts show only mild and diffuse [^18^F]FDG uptake without focal areas. These findings represent the most reliable parameters to rule out an infection, even in early post-procedural phases (1–4 months). Therefore, in patients with low probability of VGEI, [^18^F]FDG PET/CT can also be performed immediately after EVAR. Further studies exploring the differences in visual and semi-quantitative analyses between non-infected and infected grafts are warranted.

## Figures and Tables

**Figure 1 diagnostics-13-00409-f001:**
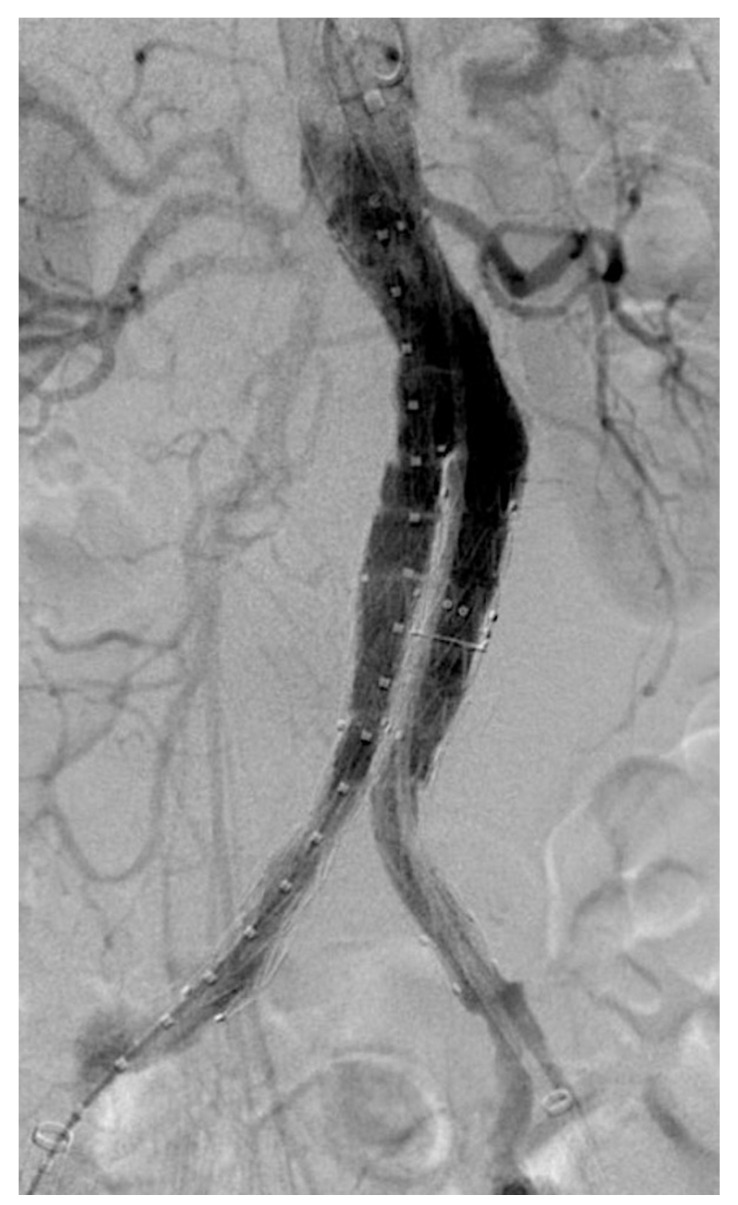
Intraoperative angiography showing complete aneurysm exclusion after Medtronic^®^ Endurant implantation.

**Figure 2 diagnostics-13-00409-f002:**
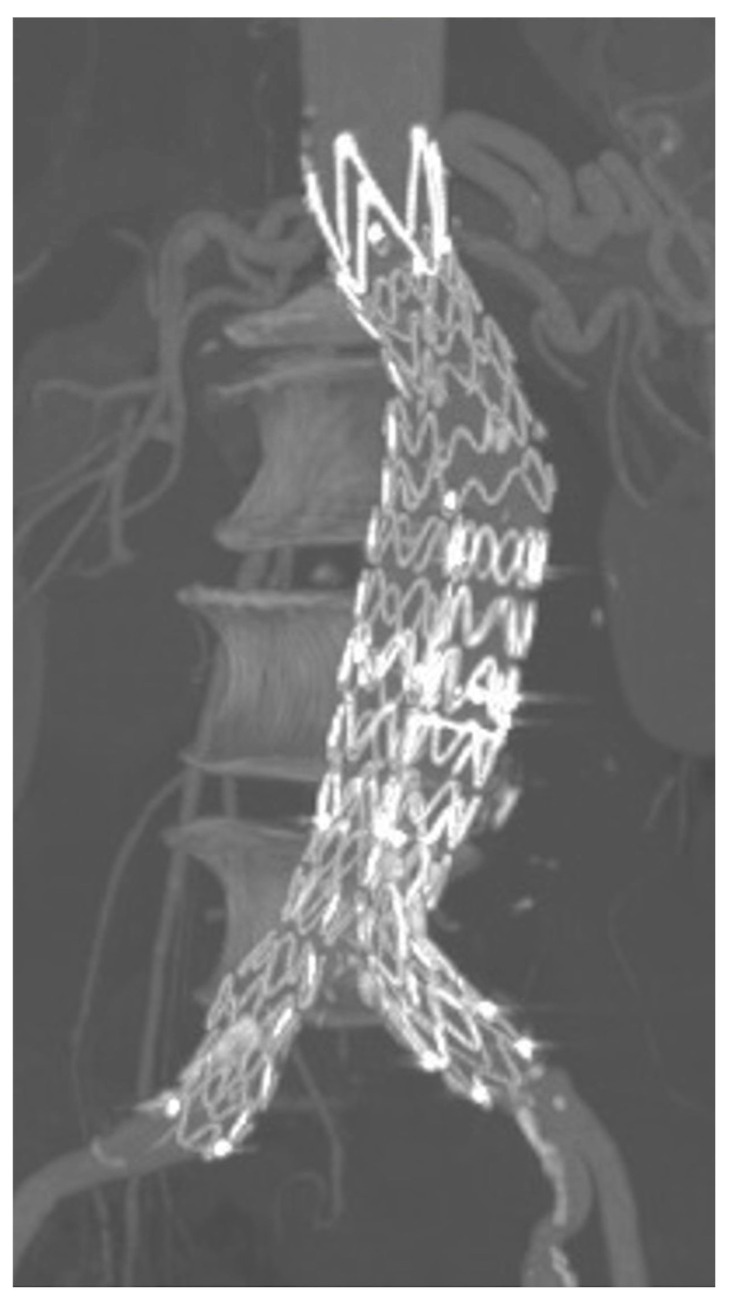
Multiplanar CTA reconstruction confirming good technical success and complete aneurysm exclusion during follow-up.

**Figure 3 diagnostics-13-00409-f003:**
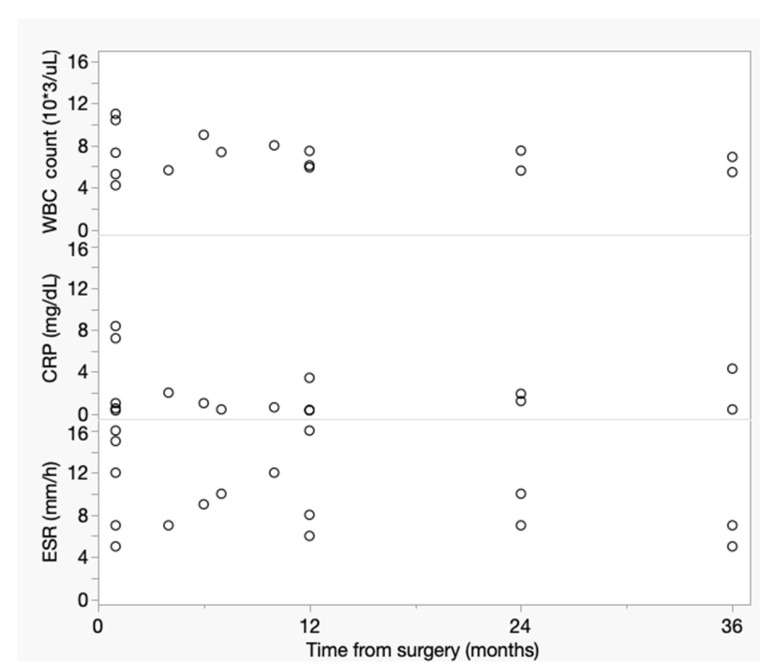
Relationship between WBC count (ρ = −0.15; 95% CI: −0.60 to 0.37; *p* = 0.57), CRP (ρ = −0.07; 95% CI: −0.54 to 0.44; *p* = 0.81) and ESR (ρ = −0.30; 95% CI: −0.70 to 0.23; *p* = 0.26) values and time from EVAR. None of these inflammatory markers was correlated with time elapsed from the endovascular procedure.

**Figure 4 diagnostics-13-00409-f004:**
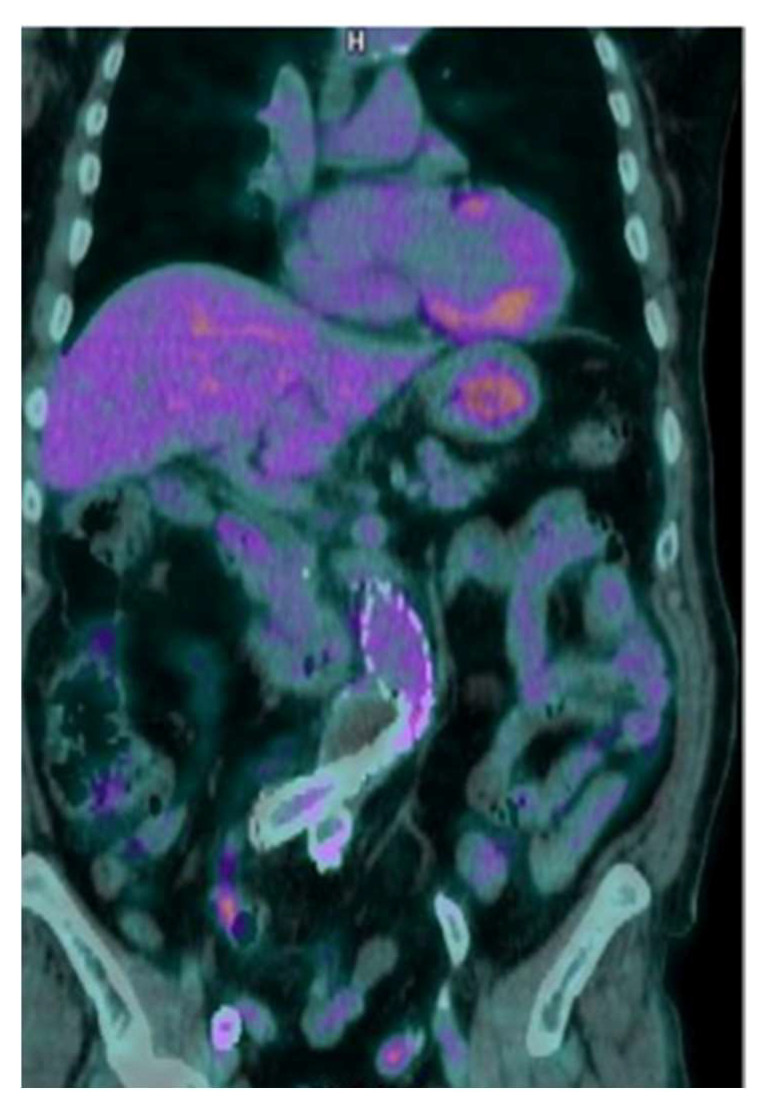
Coronal view of a negative [^18^F]FDG PET/CT, showing mild and diffuse [^18^F]FDG uptake along the graft.

**Figure 5 diagnostics-13-00409-f005:**
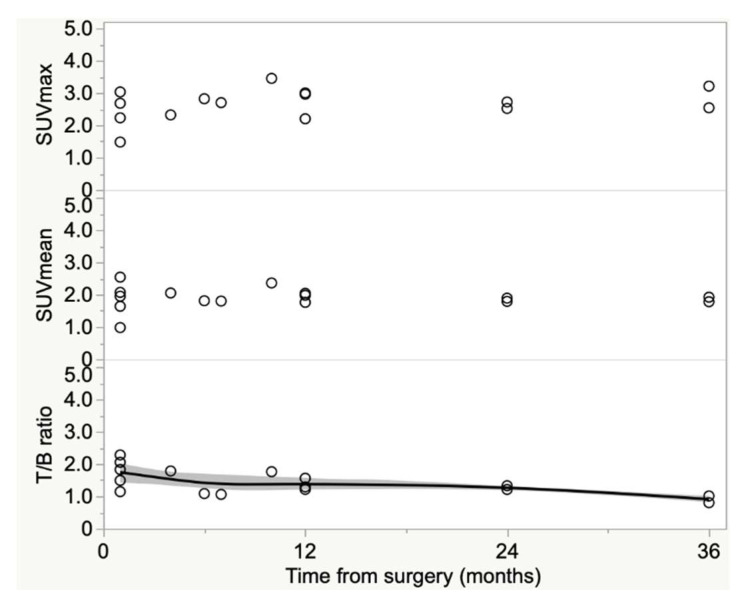
SUVmax, SUVmean and T/B ratios’ trend according to time from EVAR. Only T/B ratios showed a declining trend with increasing time (ρ = −0.61, 95% CI: (−0.85 to −0.16), *p* = 0.01.

**Table 1 diagnostics-13-00409-t001:** Characteristics of studied population.

Parameter	Absolute Frequency (%)
Gender (male/female)	13 (81.25)/3 (18.75)
Smoke (yes/no)	6 (37.50)/10 (62.50)
Hypertension (yes/no)	11 (68.75)/5 (31.25)
Dyslipidemia (yes/no)	7 (43.75)/9 (56.25)
Diabetes (yes/no)	3 (18.75)/13 (81.25)
Previous cardiac attack (yes/no)	6 (37.50)/10 (62.50)
**Parameter**	**Mean ± SD–(95% CI)**
Age (years)	71.81 ± 6.23–(68.49–75.13)
WBC count (10^3^/µL)	7.07 ± 1.87–(6.07–8.06)
CRP (mg/dL)	2.08 ± 2.52–(0.73–3.42)
ESR (mm/h)	9.50 ± 3.72–(7.51–11.48)
SUVmax	2.63 ± 0.48–(2.38–2.88)
SUVmean	1.90 ± 0.33–(1.72–2.08)
T/B ratio	1.43 ± 0.41–(1.21–1.65)

WBC: white blood cells, CRP: C-reactive protein, ESR: erythrocyte sedimentation rate, SUVmax: maximum standardized uptake value, SUVmean: mean standardized uptake value, T/B: target/background.

**Table 2 diagnostics-13-00409-t002:** Patient’s description.

Pt.	Time from Surgery (Months)	ESR (mm/h)	WBC Count (10^3^/uL)	CRP (mg/dl)	SUVmax	SUVmean	T/B Ratio
1	36	5	6.90	0.40	3.21	1.78	0.80
2	36	7	5.45	4.30	2.54	1.92	1.01
3	24	7	7.50	1.90	2.72	1.79	1.33
4	24	10	5.60	1.20	2.52	1.89	1.21
5	12	16	5.90	0.36	2.96	2.04	1.29
6	12	8	6.10	3.42	3.00	1.98	1.56
7	12	6	7.47	0.30	2.20	1.76	1.21
8	10	12	8.00	0.60	3.45	2.36	1.76
9	6	9	9.00	1.00	2.82	1.81	1.08
10	7	10	7.36	0.40	2.70	1.80	1.06
11	4	7	5.64	2.00	2.32	2.05	1.78
12	1	5	7.30	0.50	1.48	0.98	2.28
13	1	12	11.00	1.00	2.23	1.95	2.05
14	1	16	5.26	7.20	3.03	2.07	1.14
15	1	15	10.40	8.37	2.23	1.64	1.49
16	1	7	4.20	0.30	2.68	2.54	1.83

Pt: patient number; ESR: erythrocyte sedimentation rate, WBC: white blood cells, CRP: C-reactive protein, SUVmax: maximum standardized uptake value, SUVmean: mean standardized uptake value, T/B: target/background ratio.

## Data Availability

Data are available (on request) from the corresponding authors (A.S.).

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
