# Peer review of "[18F]FDG Uptake in Non-Infected Endovascular Grafts: A Retrospective Study"

_diagnostics, 2023, doi:10.3390/diagnostics13030409_

Round 1
Reviewer 1 Report
many typos: most common aneurysm and not aneurism!!!
severe linguistic revision from native speaker is required!
why did you include only oncological patients? it would be more interesting and valuable to have a control group without malignancy
title is misleading. you are only describe 18 cases of oncological patients treated by endovascular means with the Endurant stent graft
Line 22: surgical implant of abdominal aorta Endurant endografts and line 31: endovascular implant
EVAR is an endovascular procedure and can be done even percutaneously and should not be considered as a surgical procedure
Line 39: EVAR is indicated for abdominal aortic aneurysms and TEVAR for thoracic pathologies. Please correct the sentence appropriate.
references i, ii - EVAR short term superiority in comparison to open surgery. Please include also the european guidilines supporting this.
line 57: two commas
Line 58-59: word physiological should be deleted, replace post-surgical with post-procedural
please explain the oncological exclusion criteria in order to be able to be understood even from vacsular surgeons
please provide more information about the oncological stage of each patient at the point of endovascular procedure, like active malignancy etc
line 80: is still yellow marked
Figure 1: it is not possible to see the whole aneurysm therefore not able to state complete aneurysm exclusion.
Figure 2: same comment to figure 1.
Paragraph 2.3.: there is no worth to describe thoroughly the IFUs of Endurant. Please delete it.
Ethical requirements: please place this paragraph before the preoperative work-up, intervention etc
Table 1:
CV diseases: what does this diagnosis include?
Line 205: what do you mean with that?
The conclusion of the abstract does not match 10% to the conclusion of the full text. Please correct it.
Author Response
Many typos: most common aneurysm and not aneurism!!! Severe linguistic revision from native speaker is required! Thank you for your comment. The text has been carefully revised.
Why did you include only oncological patients? it would be more interesting and valuable to have a control group without malignancy. We agree with the reviewer that the inclusion of a control group of non-oncological patient would be interesting although not necessarily different from the control group we chose. Indeed, FDG PET/CT studies are generally performed for diagnosis and follow-up of malignancies, while normal subjects would have had no reason for performing a PET/CT study. Other clinical indications include the diagnosis of infection or inflammatory diseases, but the aim of this retrospective study was to assess the pattern distribution of FDG in patients with Endurant grafts but without the suspicion of infection. These are the reason why we chose to include patient with malignancies. Please note that, as explained in the manuscript, we included only patients with a very limited disease burden and with primary tumors located in distant areas from the graft. This was done in order to reduce the possible interference of high metabolic activity (due to the oncological disease) on FDG biodistribution on the graft, thus limiting sources of misinterpretation of graft’s activity.
Title is misleading. You are only describe 18 cases of oncological patients treated by endovascular means with the Endurant stent graft. Thank you for the suggestion. We modified the title accordingly.
Line 22: surgical implant of abdominal aorta Endurant endografts and line 31: endovascular implant. EVAR is an endovascular procedure and can be done even percutaneously and should not be considered as a surgical procedure. Thank you for your suggestion. We modified the text accordingly. EVAR and TEVAR are now indicated as endovascular procedures.
Line 39: EVAR is indicated for abdominal aortic aneurysms and TEVAR for thoracic pathologies. Please correct the sentence appropriate. Thank you for the suggestion. We modified the introduction accordingly.
References i, ii - EVAR short term superiority in comparison to open surgery. Please include also the european guidilines supporting this. Thank you for the suggestion. We modified the references’ list accordingly.
line 57: two commas Thank you, we amended.
Line 58-59: word physiological should be deleted, replace post-surgical with post-procedural. Ok, we changed the sentence as suggested.
Please explain the oncological exclusion criteria in order to be able to be understood even from vascular surgeons. This concept has been now better clarified in the main text as well as in the answer to your previous comment.
Please provide more information about the oncological stage of each patient at the point of endovascular procedure, like active malignancy etc... Thank you for this comment. Actually, we selected only patients that underwent to EVAR procedure before the diagnosis of cancer to avoid that the vascular surgery was somehow related to cancer itself. The five patients with solitary lung nodule represent an exception. Lung nodule was occasionally detected during the pre-operative workup, before EVAR. They, therefore, were studied by FDG PET/CT one month after EVAR for the metabolic characterization of their lung nodules. This has been now clarified also in materials and methods, thank you.
Line 80: is still yellow marked. Thank you, we removed the color
Figure 1: it is not possible to see the whole aneurysm therefore not able to state complete aneurysm exclusion. Figure 2: same comment to figure 1. Thank you for your suggestion. Both figures were accordingly modified in the revised text
Paragraph 2.3.: there is no worth to describe thoroughly the IFUs of Endurant. Please delete it. Thank you for the suggestion. We erased the entire paragraph.
Ethical requirements: please place this paragraph before the preoperative work-up, intervention etc Ok, we changed.
Table 1: CV diseases: what does this diagnosis include? We include previous cardiac attack and heart failure (now mentioned in methods and in table 1). None of the patients previously had heart failure; 6 patients had myocardial infarction.
Line 205: what do you mean with that? Thank you for the question that allows us to better specify that by comparing FDG intensity in patients with and without a given risk factor (e.g. atherosclerosis) we did not observe any significant difference. It is now better explained in the manuscript.
The conclusion of the abstract does not match 10% to the conclusion of the full text. Please correct it. Thank you for the suggestion. We modified the text of both abstract and conclusions.
Reviewer 2 Report
The authors have tried to assess the pattern of [18F] FDG uptake in a small group of 16 cancer patients who had abdominal vascular grafts specifically with Endurant® for endovascular aortic repair. The chosen patients did not show any clinically viable symptoms of infection and were studied at different time-points following the intervention.
Although the study is quite interesting and could be a valuable addition it lacks clarity required for it to relevant to the readers. I encourage the authors to work on improving the sample size and overall writing for better clarity. So please see my comments below.
General comments
1. Although, the authors claim that the FDG PET values to be a surrogate for non-infection, the given study is not powered for picking up reliable evidence for lack of infection. Note that the authors are comparing patients imaged at different timepoints with more than 30% of the patients being imaged at 1-month post-surgery and other as late as 36 months. Does it indicate that the SUV values and T/B ratio show very little changes and so we assume that these values indicate that there is no infection?
2. Were all patients given a score 2 by all the three physicians to their FDG scan?
3. In Results change the sentence to ‘All the 16 patients underwent [ 18F] FDG PET/CT study as part of the standard-of-care workup for their oncological disease.’
3. If some patients had follow-up FDG PET at 1 month and later at 12 months or 36 months, the longitudinal values maybe it would have helped to understand the nature of the uptake over time as a correlate to no infection. Would you be able to conclusively say what the cut-off for SUVmax or mean would be for non-infection related FDG uptake?
4. How does the SUV values, their ratio or T/B values compare to patients who have had infections?
5. How do you justify your choice of using SUVmean for target and background for T/B calculation?
6. In conclusion you mention non-infected vascular grafts showed homogeneous uptake. How was homogeneity determined? Mention it in the methods.
Author Response
The authors have tried to assess the pattern of [18F] FDG uptake in a small group of 16 cancer patients who had abdominal vascular grafts specifically with Endurant® for endovascular aortic repair. The chosen patients did not show any clinically viable symptoms of infection and were studied at different time-points following the intervention. Although the study is quite interesting and could be a valuable addition, it lacks clarity required for it to relevant to the readers. I encourage the authors to work on improving the sample size and overall writing for better clarity. So please see my comments below. Thank you for your comments. English style has been thoroughly revised making sentences clearer. Nevertheless, this is a retrospective study and we adopted very strict inclusion and exclusion criteria therefore, we cannot increase the sample size at this moment. We plan to do this in the next years, when the number of studied patients will increase, and we will have the possibility to confirm or not the results of present study.
General comments
Although, the authors claim that the FDG PET values to be a surrogate for non-infection, the given study is not powered for picking up reliable evidence for lack of infection. Note that the authors are comparing patients imaged at different timepoints with more than 30% of the patients being imaged at 1-month post-surgery and other as late as 36 months. Does it indicate that the SUV values and T/B ratio show very little changes and so we assume that these values indicate that there is no infection? Thank you for your relevant question. Given the nature of studied population (oncologic patients without any suspicion of infection), of course, we did not perform any microbiological nor histological confirmation of the lack of infection. Moreover, all patients perform FDG scan for other reasons and they were all negative for graft infection, so there was no reason to exclude an infective process at the time of PET/CT studies. Nevertheless, these patients were followed-up by vascular surgeons according to the standard of care and no infection was detected during a follow-up of 2 years, and we believe that it is a more than appropriate timing for excluding a VGEI. As far as the behaviour of semi-quantitative parameters is concerned, as you said, they showed very little changes over time with a declining trend for T/B ratios so, yes, we can assume that these low values, even in patients studied one month after surgery, together with the pattern of FDG biodistribution, can reliably exclude an infection. As previously stated, we plan to increase the sample size in the next years to eventually confirm these findings.
Were all patients given a score 2 by all the three physicians to their FDG scan? Absolutely yes, we now reported this information in the text.
In Results change the sentence to ‘All the 16 patients underwent [18F] FDG PET/CT study as part of the standard-of-care workup for their oncological disease.’ Ok, amended.
If some patients had follow-up FDG PET at 1 month and later at 12 months or 36 months, the longitudinal values maybe it would have helped to understand the nature of the uptake over time as a correlate to no infection. Would you be able to conclusively say what the cut-off for SUVmax or mean would be for non-infection related FDG uptake? We agree with the reviewer that assessing the metabolic activity of the graft at different time- points in the same patient would be very interesting and would strengthen our findings. Indeed, having the possibility to follow-up the same patient 2 or 3 times with FDG PET/CT would have been very informative. However, we have these longitudinal data only in 2 patients. They both performed basal FDG-PET at 1 month after EVAR and then repeated the scan 12 months (one patient) and 24 months (the other one) later for cancer follow-up (Follow-up PET scan not included in the paper). The SUV values were more or less identical. We now included, in discussion, the lack of follow-up PET/CT as a limitation of this study. Overall, we can conclude that, based on our results, the cut-off values measured in these patients are indeed indicative of absence of infection. Nevertheless, we all know very well, that the SUV value is not relevant for the diagnosis of VGEI, being the pattern of FDG uptake more important.
How does the SUV values, their ratio or T/B values compare to patients who have had infections? Thank you very much for this valuable comments. This is, indeed, an aspect we are actually investigating.
How do you justify your choice of using SUVmean for target and background for T/B calculation? As reported in discussion, the choice of background tissue and methods of calculation of T/B ratios are still matter of debate. Some authors prefer to use SUVmax graft /SUVmax background, others proposed SUVmax graft /SUVmean background and some others SUVmean graft / SUVmean background. For the specific purpose of our study, since no focal uptake was detected but only faint and diffuse pattern distribution was observed along the whole graft, SUVmax could be less representative, or even misleading, than SUVmean. Moreover, we decided to adopt the same methodology used in the study published by Keidar et al. in non infected vascular grafts (Keidar Z, Pirmisashvili N, Leiderman M, Nitecki S, Israel O. 18F-FDG uptake in noninfected prosthetic vascular grafts: incidence, patterns, and changes over time. J Nucl Med. 2014 Mar;55(3):392-5.). However, further studies are warranted to definitively assess which is the best method for T/B ratios calculation
In conclusion you mention non-infected vascular grafts showed homogeneous uptake. How was homogeneity determined? Mention it in the methods. Yes, in conclusion we used “homogeneous” as a synonym of “diffuse” (without focal uptake). We now changed it in “diffuse”. Thank you for this comment.
Round 2
Reviewer 1 Report
Thank you for the revised version.
The study is now acceptable.